# CeO_2_-Supported TiO_2_−Pt Nanorod Composites as Efficient Catalysts for CO Oxidation

**DOI:** 10.3390/molecules28041867

**Published:** 2023-02-16

**Authors:** Haiyang Wang, Ruijuan Yao, Ruiyin Zhang, Hao Ma, Jianjing Gao, Miaomiao Liang, Yuzhen Zhao, Zongcheng Miao

**Affiliations:** 1Xi’an Key Laboratory of Advanced Photo-Electronics Materials and Energy Conversion Device, School of Electronic Information, Xijing University, Xi’an 710123, China; 2School of Materials Science and Engineering, Xi’an Polytechnic University, Xi’an 710048, China; 3School of Artificial Intelligence, Optics and Electronics (iOPEN), Northwestern Polytechnical University, Xi’an 710072, China

**Keywords:** Al-Ce−Pt-TiO_2_ alloy ribbon, dealloying, CO oxidation, (TiO_2_−Pt)/CeO_2_

## Abstract

Supported Pt-based catalysts have been identified as highly selective catalysts for CO oxidation, but their potential for applications has been hampered by the high cost and scarcity of Pt metals as well as aggregation problems at relatively high temperatures. In this work, nanorod structured (TiO_2_−Pt)/CeO_2_ catalysts with the addition of 0.3 at% Pt and different atomic ratios of Ti were prepared through a combined dealloying and calcination method. XRD, XPS, SEM, TEM, and STEM measurements were used to confirm the phase composition, surface morphology, and structure of synthesized samples. After calcination treatment, Pt nanoparticles were semi-inlayed on the surface of the CeO_2_ nanorod, and TiO_2_ was highly dispersed into the catalyst system, resulting in the formation of (TiO_2_−Pt)/CeO_2_ with high specific surface area and large pore volume. The unique structure can provide more reaction path and active sites for catalytic CO oxidation, thus contributing to the generation of catalysts with high catalytic activity. The outstanding catalytic performance is ascribed to the stable structure and proper TiO_2_ doping as well as the combined effect of Pt, TiO_2_, and CeO_2_. The research results are of importance for further development of high catalytic performance nanoporous catalytic materials.

## 1. Introduction

Carbon monoxide is one of the most dangerous waste gases because of its harmful impact on the environment and high toxicity to animal and human lives. As catalytic CO oxidation is an efficient method to eliminate CO pollution under low temperature conditions, it has attracted widespread research interest in recent years [1,2]. Among them, the supported Pt-based catalysts have been widely investigated since Langmuir’s first discovery [3]. Pt-based catalysts are critical to industrial CO oxidation because of their superior catalytic activity and stable catalytic properties [4,5,6]. The catalytic mechanism of Pt catalysts has been widely investigated and the results show that the reaction generally follows Langmuir–Hinshelwood (L-H) models [7,8,9]. However, the relative high cost and scarcity of noble metals, as well as their aggregation tendency as temperature rises, have retarded their further development [10,11]. Both theoretical and experimental studies have demonstrated that combining transition metal oxides [12,13] or rare earth metal ions [14,15] with noble metals is an effective method to reduce cost while maintaining stable catalytic property, which has been widely used in fuel cell and energy conversion/storage equipment. TiO_2_, as a typical metal oxide, exhibits high oxygen storage capacity and redox properties as well as active catalytic performance by enhancing the migration rate of surface-active oxygen atoms and plays an important role in the catalysis field [16,17,18]. For example, Liou’s team [19] prepared Cu-doped TiO_2_ microsphere for catalytic CO oxidation. They think that the highly dispersed doping metals can increase the exposure of copper and TiO_2_ matrix, thus leading to the improvement of catalytic performance. However, the bulk metal oxides always show poor charge transfer ability and conductivity, which hinders their full play. Combining TiO_2_ with Pt is an effective strategy to avoid the aggregation of Pt and enhance the overall property of materials. Liu’s group [20] fabricated the Pt-Au/TiO_2_-CeO_2_ catalyst and found that the introduction of TiO_2_ into a system can improve CO oxidation by enhancing the charge transfer from Pt to Au sites. Nava’s team [21] investigated the loading amount of TiO_2_ on catalytic performance of Au/TiO_2_/SBA-15 systems and concluded that the catalyst reached the highest catalytic activity when 10 wt% TiO_2_ was added. Therefore, TiO_2_ is a good promoter in improving the catalytic performance of catalysts.

In practice, the metallic catalysts or metal–metal oxide composites are always supported on some nanostructured substrates to form heterogeneous catalysts [22]. This unique structure can allow good dispersion of noble metals and make full play use of the catalysts. It is well established that the noble catalysts supported on reducible metal oxides are more active than non-reducible oxides such as Al_2_O_3_ or SiO_2_ [23,24]. In comparison, as a unique rare metal oxide, CeO_2_ has been applied as a superior reducible supporting oxide due to its rich reservation and fast storage/release oxygen ability [25]. More importantly, the reversible Ce^3+^/Ce^4+^ redox reaction and easy generation of oxygen vacancies in CeO_2_ can contribute to the improvement in CO oxidation rate [26,27]. Previous studies also imply that the morphology and facets of CeO_2_-based nanocomposites can greatly influence the formation and migration of surface oxygen vacancies, and nanosized structured CeO_2_ materials, including nanospheres, nanorods, and nanocubes [28,29], have been synthesized. Among these structures, nanorod-shaped CeO_2_ has received a substantial amount of attention because of its potentially large surface area and abundance of oxygen vacancy defects. Li et al. [30] prepared Au cluster-CeO_2_ catalysts and concluded that the Au25 nanoclusters on CeO_2_ nanorods and nano polyhedra display higher activity than CeO_2_ nanocubes due to the difference in concentration of (O) species on ceria surface. Kwangjin An’s group [31] fabricated Pt/CeO_2_ with different morphologies and found that the Pt/CeO_2_ with cube morphology shows the best activity compared with other structured samples. It is therefore predicated that the catalytic activity of CeO_2_-based catalysts can be controlled by tuning their physicochemical properties. However, the conventional fabrication methods always require relatively high cost and complicated or time-consuming preparation processes, which limit their large-scale application.

The structure and activity of a catalyst is greatly related to the synthesis method. Compared with the traditional preparation method, dealloying is a simple and pollution-free method to fabricate three-dimensional nanoporous materials on a large-scale production basis [32]. The structure and pore size of samples can also be controlled by adjusting the dealloying temperature or composition of precursor alloys [33]. Metal oxides such as NiO [34] and CuO [35] or noble metals such as Ag [36], Au [37], and Pt [4] have been reported to be successfully supported on CeO_2_ and have displayed satisfying catalytic activity. Whereas the Pt/TiO_2_ composites supported onto CeO_2_ to improve catalytic activity has been rarely reported.

Herein, the nanorod structured (TiO_2_−Pt)/CeO_2_ catalysts with the addition of Pt and varied amount of TiO_2_ were fabricated through a combined dealloying and calcination method. The highly dispersed Pt and TiO_2_ nanoparticles are loaded onto CeO_2_ and form a nanoscale interface, which can accelerate the movement rate of electrons at the interface. The good framework structure also makes CO access catalysts more efficiently and gives full play to the role of active phases. The (0.5TiO_2_−Pt)/CeO_2_ catalyst shows optimal catalytic property of 50% and 99% at reaction temperatures as low as 55 °C and 90 °C, respectively. This work provides a new idea for preparation of high catalytic performance transition metal/CeO_2_-based catalysts for large-scale production.

## 2. Results and Discussion

### 2.1. Characterization of Catalysts

Figure 1a displays the XRD patterns of melt-spun and dealloyed Al_91.2_Ce_8_Pt_0.3_Ti_0.5_ ribbons. As observed, the melt-spun Al_91.2_Ce_8_Pt_0.3_Ti_0.5_ ribbons consisted of α-Al, Al_4_Ce and Al_92_Ce_8_ phases; after the dealloying procedure, only a new phase of CeO_x_ was detected while α-Al, Al_4_Ce, and Al_92_Ce_8_ phases disappeared, implying that most of the Al has been removed. The diffraction peaks representing Pt/Ti cannot be detected, which is ascribed to their low content and high dispersion into alloy ribbons. The XRD patterns of Al_91.4_Ce_8_Pt_0.3_Ti_0.3_, Al_91.2_Ce_8_Pt_0.3_Ti_0.5_, and Al_91_Ce_8_Pt_0.3_Ti_0.7_ melt-spun ribbons after dealloying and calcination treatments are displayed in Figure 1b. The diffraction at 28.5°, 32.9°, 47.4°, 56.2°, 69.2°, and 76.7° corresponded to the (111), (200), (220), (311), (400), and (331) planes of cubic CeO_2_ (PDF#89-8436), respectively; the weak diffraction peak at 41^o^ representing Pt was also discovered while no peaks related to Ti was found. However, the content of Al, Ce, Pt, and Ti in the (0.5TiO_2_-Pd)/CeO_2_ catalyst obtained from Al_91.2_Ce_8_Pt_0.3_Ti_0.5_ melt-spun ribbon is 3.81 at%, 90.14 at%, 1.66 at%, and 4.4 at%, respectively, as shown in the EDS spectrum in Appendix A, demonstrating that Pt and Ti have been added into Al-Ce precursor alloys successfully.

To further confirm the chemical state of Pt, Ti, and Ce, XPS characterization of (0.5TiO_2_−Pt)/CeO_2_ is conducted with results shown in Figure 2. The Ce 3d spectrum displayed in Figure 2a reveals that the sample exhibits both Ce^4+^ and Ce^3+^ ions. The five peaks at 881.9 eV, 888.3 eV, 897.7 eV, 900.4 eV, and 907.3 eV are ascribed to Ce^4+^, while the other two peaks at 885.1 eV and 903.7 eV corresponded to Ce^3+^. The existence of Ce^3+^ implies the generation of oxygen vacancies; Ce^3+^ can adsorb active oxygen at the catalytic interface, thus contributing to the formation of interfacial active center. The concentration of Ce^3+^ can be reflected from the integrated areas of the Ce^3+^ peak to the total (Ce^3+^ + Ce^4+^) peaks. As a result, the surface concentration of Ce^3+^ on the (0.5TiO_2_−Pt)/CeO_2_ catalyst is 21.58% according to the fitting calculation of the Ce 3d spectrum. For the Pt 4f spectrum in Figure 2b, the binding energies at 70.8 eV for Pt 4f_7/2_ and 73.9 eV for Pt 4f_5/2_ are assigned to metallic state platinum (Pt^0^), while the peaks at 71.9 eV and 76.4 eV corresponded to Pt^2+^ [38,39]. Likewise, the content of Pt^0^ accounts for 61.6% of the total (Pt^0^ + Pt^2+^). The Ti 2p spectrum in Figure 2c displays a Ti^4+^ binding energy, in which the two peaks at 463.6 eV and 457.8eV corresponded to Ti 2p_1/2_ and Ti 2p_3/2_, respectively [40]. Since Ti mainly existed in the form of Ti^4+^ in the product, it is deduced that TiO_2_ existed in the composite material. The O 1s spectrum in Figure 2d can be fitted to three peaks. The binding energies centered around ~529.3 eV, ~531 eV, and ~532.2 eV corresponded to lattice oxygen species (O_lat_), surface adsorbed oxygen (O_sur_), and weakly bonded specific oxygen species such as adsorbed O_2_, H_2_O, and CO_2_ (O_bon_), respectively. The active surface oxygen can be evaluated by O_sur_, and the ratio of active oxygen species for (0.5TiO_2_−Pt)/CeO_2_ is 20.8%.

Figure 3 presents the surface and cross-sectional morphologies of (TiO_2_−Pt)/CeO_2_ with different TiO_2_ content. As observed, all the three samples display a robust framework, which are composed of a nanoporous matrix with nanorods embedded in them. The nanorods pile up on each other to form rich pores among them. Notably, the slight increase in TiO_2_ content from 0.3 at% to 0.5 at% does not influence the overall morphologies of samples and only fine-tunes the arrangement of pores, as shown in Figure 3a,d,g. Moreover, the cross-sectional SEM image of (0.5TiO_2_−Pt)/CeO_2_ in Appendix A further reflects the presence of rich pores and independent arrangement of nanorods. The unique and robust nanorod-embedded matrix structure is beneficial to stabilize the overall structure of samples during the catalytic process; the existence of lots of pores distributed among matrix and nanorods can also provide more channels for reacted gas to enter and exit; therefore, the catalytic CO oxidation performance is expected to be improved.

TEM and HRTEM characterization are performed to further understand the microstructure of (TiO_2_−Pt)/CeO_2_ catalysts. As shown in the TEM images of (0.3TiO_2_−Pt)/CeO_2_, (0.5TiO_2_−Pt)/CeO_2_, and (0.7TiO_2_−Pt)/CeO_2_ presented in Figure 3b,e,h, respectively, the samples are composed of a large number of uniform nanorods with an average diameter of 10 nm, which are interconnected and stacked on each other; some dark nanoparticles with diameter of 3–5 nm on average are uniformly embedded on the surface of nanorods. These are consistent with SEM results. The corresponding HRTEM images of (0.3TiO_2_−Pt)/CeO_2_, (0.5TiO_2_−Pt)/CeO_2_, and (0.7TiO_2_−Pt)/CeO_2_ are displayed in Figure 3c,f,i, respectively. The lattice fringe with a space of 0.32 nm corresponded to the (111) plane of CeO_2_, implying the cubic structured CeO_2_ nanorod in the (111) crystal plane. The dark nanoparticles with lattice space of 0.229 nm are assigned to the (111) plane of Pt, which further indicates that Pt has been added into Al-Ce alloy successfully. However, no results related to Ti are found in TEM characterization. This may be because the calcination temperature in the (TiO_2_−Pt)/CeO_2_ system is relatively low (300 °C); CeO_2_ can inhibit the crystallization of other oxides during the calcination process under such low temperatures [41]. Our previous work also found that CeO_2_ can inhibit the crystallization of NiO; as temperature rises, the structure of NiO in the system is transformed from the amorphous state into the crystallization state [34]. Therefore, the reason why the lattice fringe related to TiO_2_ is not detected in TEM characterization may be the amorphous state of TiO_2_ in the system, which is in line with XRD results.

The distribution of elements on the surface of the CeO_2_ nanorod is further investigated via STEM mapping, with results presented in Figure 4. Figure 4a displays the SEM image of (0.5TiO_2_−Pt)/CeO_2_. For (0.5TiO_2_−Pt)/CeO_2_ obtained from Al_91.2_Ce_8_Pt_0.3_Ti_0.5_ through the dealloying and calcination processes, Pt is semi-embedded onto the surface of the CeO_2_ nanorod, while Ti is uniformly distributed into the CeO_2_ nanorod, as reflected in Figure 4b–d. Combined with XPS and STEM results, it can be concluded that Ti mainly exists as the TiO_2_ phase in the composite system; thus, the obtained composite material is named as (TiO_2_−Pt)/CeO_2_.

The specific surface area, pore size distribution, and pore volume of (TiO_2_−Pt)/CeO_2_ composite materials with varied TiO_2_ proportions are measured via the N_2_ adsorption-desorption test, with results displayed in Figure 5. The isotherms of three catalysts belong to type IV and possess H3 hysteresis loops at relative pressure of 0.7–1.0 P/P_0_ according to the IUPAC classification (Figure 5a), indicating the mesoporous structure of (TiO_2_−Pt)/CeO_2_ [42]. The BET surface area of (0.3TiO_2_−Pt)/CeO_2_, (0.5TiO_2_−Pt)/CeO_2_, and (0.7TiO_2_−Pt)/CeO_2_ is 101.88, 108.88, and 110.11 m^2^ g^−1^, respectively, while their corresponding pore size is centered at 14.36, 12.71, and 13.58 nm, and pore volume is 0.36, 0.37, and 0.35 cm^3^ g^−1^, respectively, as displayed in the BHJ pore size distribution curves in Figure 5b. Obviously, the three catalysts possess similar results in specific surface area and pore size distribution, which illustrates that the variation in the amount of Pt and TiO_2_ does not influence the physical structure of materials significantly, nor their mesoporous properties. In contrast, (0.5TiO_2_−Pt)/CeO_2_ has higher specific surface area, larger pore volume, and smaller porosity, which is beneficial for gas penetration during the catalytic process by providing more reaction paths and active sites for catalytic CO oxidation, and thus improving its catalytic performance.

Raman spectroscopy measurement is conducted to understand the structural phase changes of (TiO_2_−Pt)/CeO_2_ catalysts. In Figure 6, the weak peaks of Raman shift around 306 and 534 cm^−1^ indicate the existence of anatase TiO_2_; the appearance of new and broad peaks around 269 cm^−1^ is attributed to co-doping of Pt [43,44]. Moreover, compared with Raman peaks of pure CeO_2_ in Appendix A, the diffraction peak is shifted from 459 cm^−1^ to 439 cm^−1^, which is ascribed to the formation of more grain boundaries after the addition of TiO_2_ and Pt nanoparticles. It is expected that the Pt and TiO_2_ nanoparticles that are highly dispersed on CeO_2_ nanorods can cause a large number of defects including oxygen vacancies, grain boundaries, and dislocations, which are helpful for improvement in catalytic activity of catalysts.

### 2.2. Catalytic Performance

Figure 7 presents the catalytic CO oxidation performance of (TiO_2_−Pt)/CeO_2_ catalysts. For Pt_0.3_/CeO_2_ without the addition of TiO_2_, the temperature for 50% CO conversion (T_50_) and 99% CO conversion (T_99_) is 91°C and 113 °C, respectively, which is much higher than that of the CeO_2_ matrix (T_50_ = 235 °C, T_99_ = 320 °C), as observed in Appendix A. The catalytic activity is greatly improved after the addition of TiO_2_. The T_50_ and T_99_ of (0.3TiO_2_−Pt)/CeO_2_ is 65 °C and 110 °C, respectively, when 0.3 at% Ti is added into alloy system. As Ti content increases to 0.5 at%, the catalytic activity reaches the optimum with a T_50_ and T_99_ decrease to 55 °C and 90 °C, respectively; on further increasing Ti content to 0.7 at%, catalytic performance decreases with T_50_ and T_99_ of 65 °C and 100 °C, respectively, as displayed in Figure 7a. The influence of calcination temperature on catalytic property of the (0.5TiO_2_−Pt)/CeO_2_ catalyst is shown in Figure 7b, in which the T_99_ of (0.5TiO_2_−Pt)/CeO_2_ without calcination treatment, calcined at 200 °C, 300 °C, 400 °C, and 500 °C is 120 °C, 110 °C, 90 °C, 100 °C, and 120 °C, respectively. The catalytic performance of (0.5TiO_2_−Pt)/CeO_2_ was stable after three repeated tests (Appendix A), implying good reusability of (0.5TiO_2_−Pt)/CeO_2_. The catalytic activity of (0.5TiO_2_−Pt)/CeO_2_ also surpasses the state-of-the-art TiO_2_/CeO_2_-based catalysts reported in the literature, as shown in Table 1 [22,45,46,47,48], indicating its superior catalytic property. It is clearly observed that the catalytic activity is improved as calcination temperature increases from room temperature to 300 °C, which is reduced as calcination temperature further increases. The (0.5TiO_2_−Pt)/CeO_2_ exhibits optimum catalytic performance after calcination at 300 °C. Furthermore, the addition of Ti into the Pt-CeO_2_ catalytic system can partly make up for the deficiency of the single precious metal Pt and realize the purpose of the experiment.

The catalytic performance of (0.5TiO_2_−Pt)/CeO_2_ as a function of flow rate at 70 °C is detected, with corresponding catalytic activities shown in Figure 8a. As the total gas flow rate increases from 40 to 120 mL min^−1^, the CO conversion decreases from 97% to 58%. It can be also clearly detected that the reaction rate is positively related to flow rate. Figure 8b further explores the influence of O_2_ concentration in feed gas on catalytic performance of (0.5TiO_2_−Pt)/CeO_2_. The test temperature is kept at 90 °C with a flow rate of 100 mL min^−1^. The CO conversion rate can reach 99% as 10% O_2_ is initially infused into the system thanks to the sufficient O_2_ environment; CO conversion rate is reduced first and then kept stable at 10% when O_2_ supply is suddenly decreased to zero, which may be ascribed to the existence of surface lattice oxygen that can migrate to active sites and combine with adsorbed CO to form oxygen vacancies. However, CO conversion rate increases in poor oxygen conditions (0.3–5% O_2_) and then recovers to initial 99% value and stays unchanged when O_2_ is resupplied into feed gas, implying the superior catalytic CO oxidation property of (0.5TiO_2_−Pt)/CeO_2_.

The long-term stability of the (0.5TiO_2_−Pt)/CeO_2_ catalyst is also evaluated to investigate its practical application potential, as shown in Figure 9a. The (0.5TiO_2_−Pt)/CeO_2_ catalyst exhibits above 95% CO conversion under mixed atmosphere (1% CO, 10% O_2_, 89% N_2_) and is stable without deterioration after successive reaction of 55 h, indicating outstanding catalytic activity of the nanorod-shaped (0.5TiO_2_−Pt)/CeO_2_ catalyst. The outstanding catalytic performance of the (TiO_2_−Pt)/CeO_2_ catalyst can be attributed to the unique structure and phase composition. The existence of Ce^3+^ on catalytic interface can adsorb active oxygen, which is conducive to the formation of the interfacial active center; highly dispersed TiO_2_ can accelerate the migration rate of active oxygen species on the surface of CeO_2_ so that the oxygen atoms can react with activated CO to form CO_2_ [34], as reflected in the mechanism diagram in Figure 9b. The introduction of Pt nanoparticles and highly dispersed TiO_2_ can form a large number of nanoscale interfaces, which greatly promotes the movement of electrons at the interface. The electrons can not only activate the CO gas adsorbed by noble metals quickly but also accelerate the dissociation of generated CO_2_ on the catalyst surface, thus ultimately making the reaction rate increase. In addition, the robust framework structure provides a place for catalysts to contact harmful gases effectively; it also stimulates the effect of noble metals that are loaded on the CeO_2_ structure and inhibits the agglomeration or growth of loaded nanoparticles during heating or catalytic processes, guaranteeing the high catalytic stability of the catalysts.

## 3. Materials and Methods

### 3.1. Material Preparation

The Al_92_Ce_8_, Al_91.7_Ce_8_Pt_0.3_, Al_91.4_Ce_8_Pt_0.3_Ti_0.3_, Al_91.2_Ce_8_Pt_0.3_Ti_0.5_, and Al_91_Ce_8_Pt_0.3_Ti_0.7_ alloys were achieved from pure Al, Ce, Pt, and Pd through the arc-melting method under high-purity Ar atmosphere. After being remelted and solidified, the Al-Ce−Pt-Ti alloy ribbons with 4–6 mm width and 40–70 μm thickness were prepared. The quenched alloy ribbons were dealloyed in 20 wt% NaOH aqueous solution at room temperature for 2 h until no obvious bubbles were generated and most of Al were removed. After this, the samples were then further corroded at 80 °C for 10 h. Finally, after cleaning and drying, the dealloyed samples were calcined at 200–500 °C for 2 h under pure O_2_ environment.

### 3.2. Characterization

X-ray diffraction patterns were collected on Bruker D8 Advance to analysis phase composition. Field emission scanning electron microscopy (FESEM, JEOL, JSM-7000F) and high-resolution transmission electron microscopy (HRTEM, JEOL, JEM-2100) were employed to characterize surface morphologies and microstructures. A scanning transmission electron microscope (STEM, FEI-200) equipped with an Oxford Instruments EDS spectrometer was utilized to conduct EDS analysis and mapping. X-ray photoelectron spectroscopy (XPS) was performed on ESCALAB Xi+ to confirm element composition and valence state. Nitrogen sorption was tested on Micromeritics ASAP 2020 at 77 K, and the Barrett–Joyner–Halenda algorithm was adopted to evaluate pore size and pore volume. Raman spectra were collected on an HR 800 fully automatic laser Raman spectrometer.

### 3.3. Catalytic Evaluations

The catalytic activity was detected in a tubular reactor at atmospheric pressure. A 100 mg sample was added to the reactor and fixed with quartz wool. The mixed reaction gas consisting of 1% CO, 10% O_2_, and 89% N_2_ (volume fraction) was entered into the test system at a flow rate of 100 mL min^−1^ (space velocity 60,000 h^−1^). The inflowed and outflowed gases were collected using an Anglit 7890B gas chromatograph equipped with a hydrogen flame detector (FID). The CO conversion was determined by:
(1)
$$\mathrm{CO}\;\mathrm{conversion}=\frac{C_{in}-C_{out}}{C_{in}}\times 100\%$$

where *C_in_* and *C_out_* stand for the concentration of the CO inlet and outlet of the reactor, respectively.

## 4. Conclusions

In conclusion, the nanorod structured (TiO_2_−Pt)/CeO_2_ catalysts are fabricated via the combined dealloying and calcination method. SEM, TEM, and STEM measurements imply that the Pt nanoparticles were semi-inlayed on the surface of the CeO_2_ nanorod, while TiO_2_ were highly dispersed into the catalyst system. By rationally adjusting the proportion of TiO_2_ in the system, the obtained (0.5TiO_2_−Pt)/CeO_2_ displays unique nanorod structure and large pore volume, which contributes to exceptional catalytic activity with T_50_ and T_99_ temperature as low as 55 °C and 90 °C, respectively. It is considered that the stable structure, proper TiO_2_ doping, and jointed effect of Pt and TiO_2_ as well as rich nanopores contribute to the enhanced catalytic performance of (TiO_2_−Pt)/CeO_2_ catalysts. This work provides a new idea and facile strategy for the fabrication of noble metal/metal oxide composites with high catalytic performance.

## Figures and Tables

**Figure 1 molecules-28-01867-f001:**
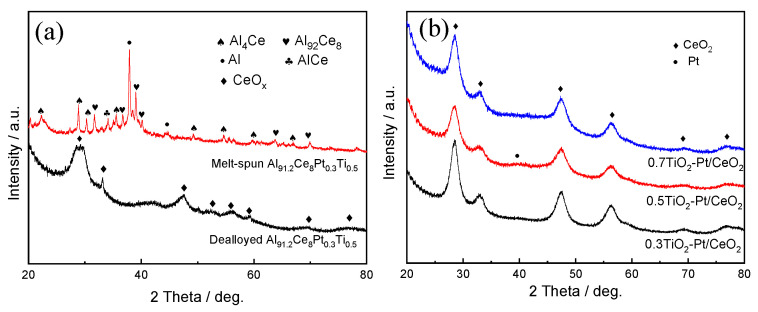
The XRD patterns of (**a**) melt-spun and dealloyed Al_91.2_Ce_8_Pt_0.3_Ti_0.5_ ribbons; (**b**) (xTiO_2_−Pt_0.3_)/CeO_2_ (X = 0.3, 0.5, 0.7) calcined at 300 °C.

**Figure 2 molecules-28-01867-f002:**
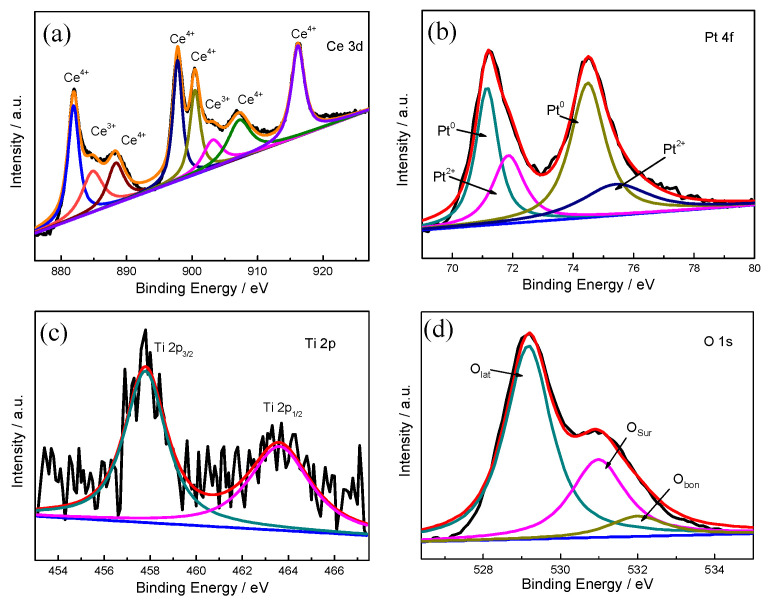
XPS spectra of (**a**) Ce 3d, (**b**) Pt 4f, (**c**) Ti 2p, and (**d**) O 1s of the (0.5TiO_2_−Pt)/CeO_2_ catalyst.

**Figure 3 molecules-28-01867-f003:**
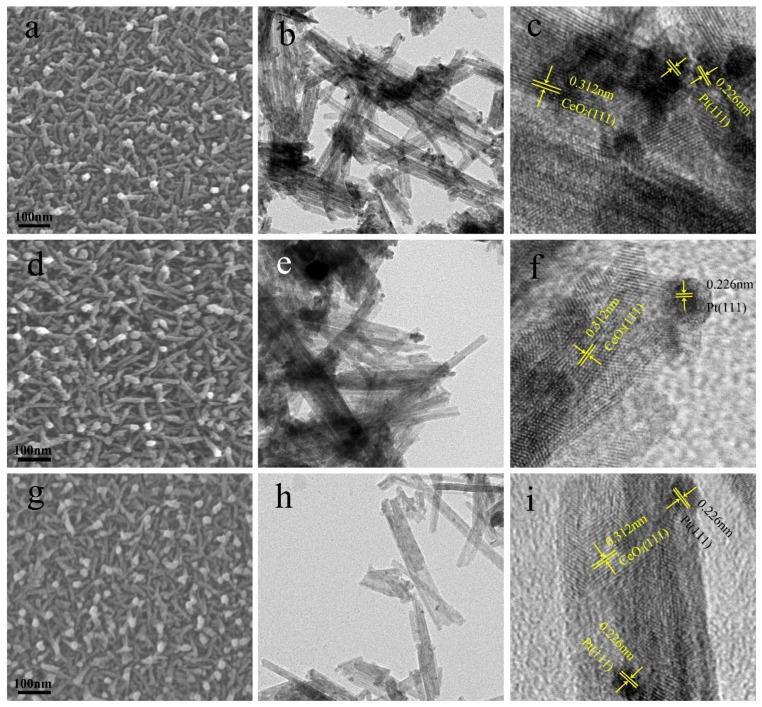
The SEM images of (**a**) (0.3TiO_2_−Pt)/CeO_2_, (**d**) (0.5TiO_2_−Pt)/CeO_2_, and (**g**) (0.7TiO_2_−Pt)/CeO_2_; the TEM and HRTEM images of (**b**,**c**) (0.3TiO_2_−Pt)/CeO_2_, (**e**,**f**) (0.5TiO_2_−Pt)/CeO_2_, and (**h**,**i**) (0.7TiO_2_−Pt)/CeO_2_.

**Figure 4 molecules-28-01867-f004:**
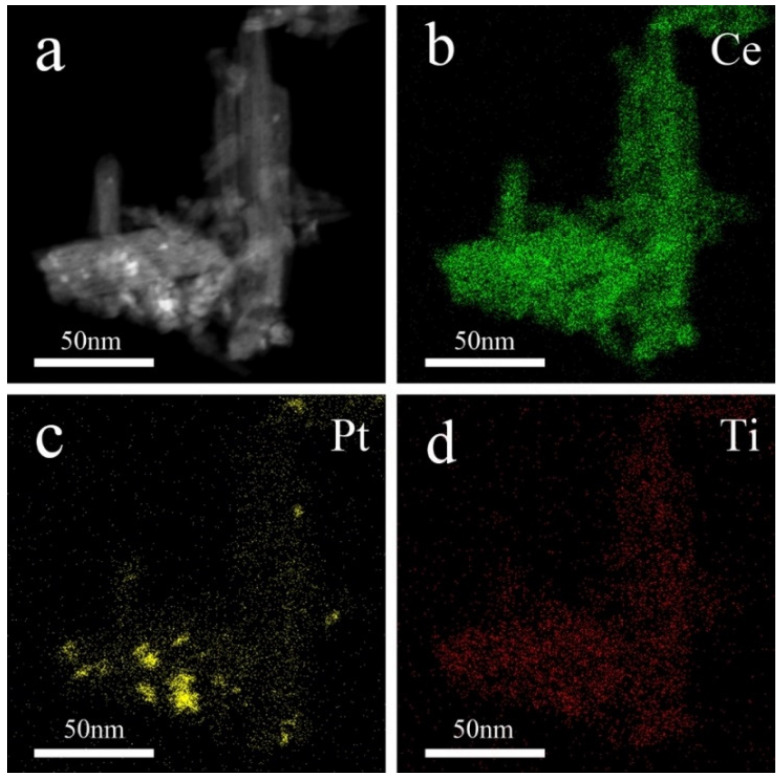
The STEM image of (**a**) (0.5TiO_2_−Pt)/CeO_2_ and element mapping of (**b**) Ce, (**c**) Pt, and (**d**) Ti.

**Figure 5 molecules-28-01867-f005:**
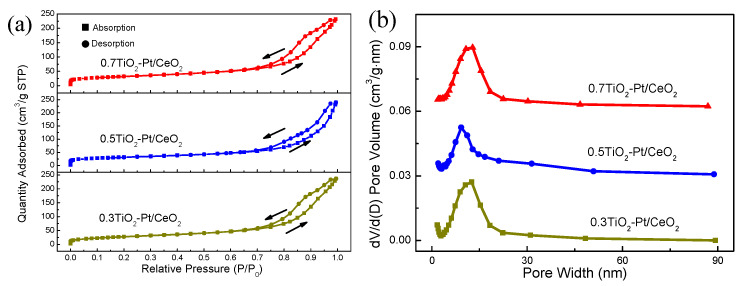
(**a**) Nitrogen adsorption–desorption isotherms and (**b**) the BHJ pore size distribution of (TiO_2_−Pt)/CeO_2_.

**Figure 6 molecules-28-01867-f006:**
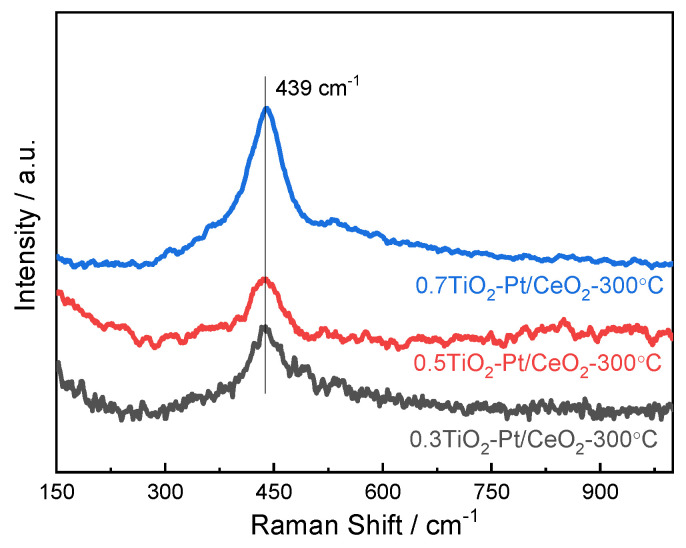
Raman spectra of (TiO_2_−Pt)/CeO_2_ catalysts.

**Figure 7 molecules-28-01867-f007:**
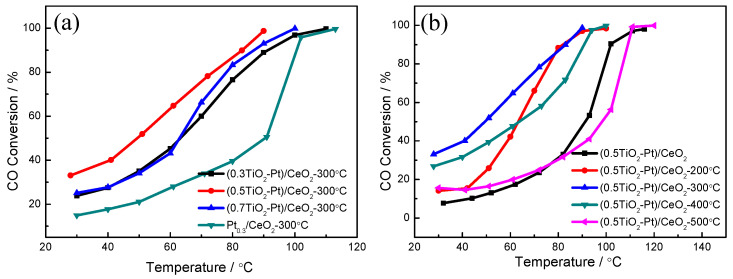
(**a**) The catalytic performance of Pt_0.3_/CeO_2_, (0.3TiO_2_−Pt)/CeO_2_, (0.5TiO_2_−Pt)/CeO_2_, (0.7TiO_2_−Pt)/CeO_2_ catalysts; (**b**) the (0.5TiO_2_−Pt)/CeO_2_ catalyst obtained at different calcination temperatures.

**Figure 8 molecules-28-01867-f008:**
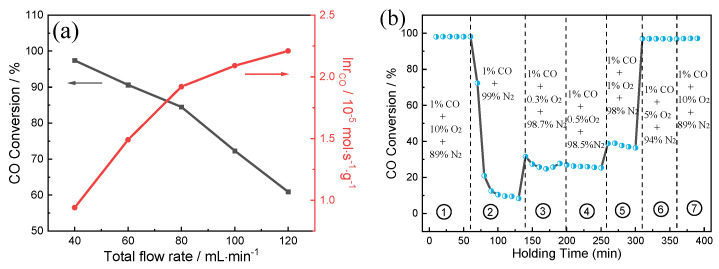
(**a**) Catalytic activity of (0.5TiO_2_−Pt)/CeO_2_ under different space velocities at 70 °C. The measurement was performed using 100 mg of the catalyst with a mixed gas of 1% CO, 10% O_2_, and rest N_2_ at a flow rate ranging from 40 to 120 mL min^−1^. (**b**) Catalytic performance under varied oxygen concentrations.

**Figure 9 molecules-28-01867-f009:**
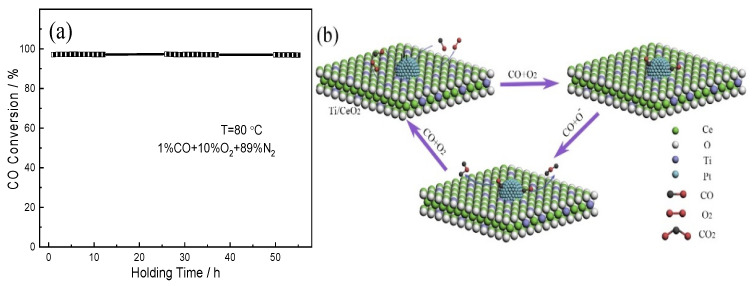
(**a**) The long-term stability of the (0.5TiO_2_−Pt)/CeO_2_ catalyst; (**b**) a possible mechanism for reaction of CO on the (TiO_2_−Pt)/CeO_2_ catalyst.

**Table 1 molecules-28-01867-t001:** Comparison on catalytic performance of (0.5TiO_2_−Pt)/CeO_2_ with previous reports.

Sample	Preparation Method	Test Condition	T_50_ (°C)	T_99_ (°C)	Reference
Pd/Pr-CeO_2_-5%	Hydrothermal synthesis	1% CO, 99% dry air	/	160	[22]
Pt/CeO_2_	Electrostatic Adsorption	1% CO, 20% O_2_, He balance	140	/	[45]
Ir/CeO_2_	wet chemical reduction	1% CO	/	110	[46]
Au/TiO_2_-S	Deposition-precipation method	2.0% CO, 8% O_2_, He balance	/	20	[47]
Co_3_O_4_@CeO_2_	Hydrothermal method	1% CO, 99% air	/	160	[48]
(0.5TiO_2_−Pt)/CeO_2_	Dealloying and calcination	1% CO, 10% O_2_, 89% N_2_	55	90	This work

## Data Availability

The raw data are available from the corresponding author upon reasonable request.

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
