# Peer review of "CeO2-Supported TiO2−Pt Nanorod Composites as Efficient Catalysts for CO Oxidation"

_molecules, 2023, doi:10.3390/molecules28041867_

Round 1

Reviewer 1 Report

Comments and Suggestions for Authors

1.      Please insert superiority of the prepared catalyst over some reported catalysts for a model reaction in a Table.

2.      What is the results of oxidation reaction in the presence of only TiO2 or CeO2 or without the catalyst? Please clarify in the manuscript.

3.      What is the results about the reusability of the synthesized catalyst? Please clarify in the text.

4.      What is the role of each part of the catalyst in oxidation of CO? Please explain in detail with some experimental tests and literature references.

5.      The intensity of the Raman peaks are very low. Please check the Raman spectra and improve them.

Author Response

Dear Reviewer:

Thank you for your comments concerning our manuscript entitled “Enhanced catalytic performance of CeO2 supported TiO2-Pt nanorod in CO oxidation” (ID: molecules-2198344). Those comments are valuable and very helpful for revising and improving the manuscript, as well as the important guiding significance to our researches. According to the reviewers’ detailed suggestions, we have made a careful revision on the original manuscript. The revised are marked in yellow in the manuscript. The main corrections and the responds to the reviewers’ comments are as follows.

Response to reviewers’ comments:

Referee #1

Comment 1: Please insert superiority of the prepared catalyst over some reported catalysts for a model reaction in a Table.

Response: Thanks for reviewer’s kind suggestion very much. We have compared the catalytic performance of (0.5TiO2-Pt)/CeO2 with the state of art TiO2/CeO2-based catalysts reported in literature, as shown in Table S1, indicating its superior catalytic property.

Comment 2: What is the results of oxidation reaction in the presence of only TiO2 or CeO2 or without the catalyst? Please clarify in the manuscript.

Response: Thanks for reviewer’s question. We have tested the oxidation reaction of pure CeO2, and concluded that the T50 and T99 of pure CeO2 is 235 °C, 320 °C, respectively, as observed in Figure S4.

Comment 3: What is the results about the reusability of the synthesized catalyst? Please clarify in the text.

Response: Thanks for reviewer’s question. The catalytic performance of (0.5TiO2-Pt)/CeO2 keeps stable after three repeated tests, implying good reusability of (0.5TiO2-Pt)/CeO2, as provided in Figure S5.

Comment 4: What is the role of each part of the catalyst in oxidation of CO? Please explain in detail with some experimental tests and literature references.

Response: Thanks for reviewer’s question. We think the outstanding catalytic performance of (TiO2-Pt)/CeO2 catalyst can be attributed to the unique structure and phase composition. The existence of Ce3+ on catalytic interface can adsorb active oxygen, which is conducive to the formation of interfacial active center; highly dispersed TiO2 can accelerate the migration rate of active oxygen species on the surface of CeO2 so that the oxygen atoms can react with activated CO to form CO2, which can be demonstrated from the reported CeO2 and TiO2 based literatures, like Nanotechnology 28(4) (2017) 045602, Journal of Hazardous Materials 403 (2021) 123630.

Comment 5: The intensity of the Raman peaks are very low. Please check the Raman spectra and improve them.

Response: Thanks for reviewer’s question. We have retested the Raman characterization and reanalyzed the raman data, see Fig. 6.

We tried our best to improve the manuscript and made some changes in the manuscript. These changes will not influence the content and frame work of the manuscript. And here we did not list the changes but marked in blue in the revised. We appreciate Editors/Reviewers’ warm work earnestly, and hope that the correction will meet with approval. Once again, thank you very much for your comments and suggestions.  Best wishes,

 Sincerely yours!

Reviewer 2 Report

Comments and Suggestions for Authors

1.      Abstract, the authors should give the background of this study, for example, the major challenge and problem of catalytic CO oxidation should be stated.

2.      Introduction Section should be revised to add some more strong points for the motivation of this work, and to summarize the recent advances and the main challenge of this field.

3.      I suggest the authors supplement some characterizations about material redox property, such as H2 TPR, or some reaction mechanism tests, such as TPD, in situ FTIR, or some theory calculations.

4.      Section 2.2, the discussions about performance and should be revised, and combined with previous characterization result.

5.      Fig. 8b, was that a DFT result? Did any characterization and result can support this figure?

Author Response

Dear Reviewer:

Thank you for your comments concerning our manuscript entitled “Enhanced catalytic performance of CeO2 supported TiO2-Pt nanorod in CO oxidation” (ID: molecules-2198344). Those comments are valuable and very helpful for revising and improving the manuscript, as well as the important guiding significance to our researches. According to the reviewers’ detailed suggestions, we have made a careful revision on the original manuscript. The revised are marked in yellow in the manuscript. The main corrections and the responds to the reviewers’ comments are as follows.

Response to reviewers’ comments:

Referee #2

Comment 1: Abstract, the authors should give the background of this study, for example, the major challenge and problem of catalytic CO oxidation should be stated.

Response: Thanks for reviewer’s kind suggestion very much. We have added this part of content in Abstract part. “Supported Pt-based catalysts have been identified as highly selective catalysts for CO oxidation, but their potential for applications has been hampered by the high cost and scarcity of Pt metals as well as aggregation problems at relative high temperatures”.

Comment 2: Introduction Section should be revised to add some more strong points for the motivation of this work, and to summarize the recent advances and the main challenge of this field.

Response: Thanks for reviewer’s kind suggestion. We have revised these points and also supplied the recent advances of related works, like “Liou’s team (J Hazard Mater 403 (2021) 123630) prepared Cu-doped TiO2 microsphere for catalytic CO oxidation. They think the highly dispersed doping metals can increase the exposure of copper and TiO2 matrix, thus benefiting for improvement of catalytic performance. However, the bulk metal oxides always show poor charge transfer ability and conductivity, which hinders their full play. Combing TiO2 with Pt is an effective strategy to avoid the aggregation of Pt and enhance the overall property of materials.” Please see the highlighted contents in Introduction part.

Comment 3: I suggest the authors supplement some characterizations about material redox property, such as H2 TPR, or some reaction mechanism tests, such as TPD, in situ FTIR, or some theory calculations.

Response: Thanks for reviewer’s question. We have added the characterizations about the catalytic activity of (0.5TiO2-Pt)/CeO2 under different space velocities as well as the catalytic performance under varied oxygen concentrations, see Fig. 8. The catalytic performance of (0.5TiO2-Pt)/CeO2 as a function of flow rate at 70 °C is detected, with corresponding shown in Fig. 8a. As the total gas flow rate increases from 40 to 120 ml min-1, the CO conversion decreases from 97% to 58%. It can be also clearly detected that the reaction rate is positive related to flow rate. Fig. 8b further explores the influence of O2 concentration in feed gas on catalytic performance of (0.5TiO2-Pt)/CeO2. The test temperature is kept at 90 °C with flow rate100 mL min-1. The CO conversion rate can reach 99% as 10% O2 is initially infused into the system thanks to the sufficient O2 environment; CO conversion rate is reduced first and then kept stable at 10% when O2 supply is suddenly decreased to zero, which may be ascribed to the existence of surface lattice oxygen that can migrate to active sites and combine with adsorbed CO to form oxygen vacancies. However, CO conversion rate increases in poor oxygen condition (0.3-5% O2) and then recovers to initial 99% value and keeps unchanged when O2 is resupplied into feed gas, implying superior catalytic CO oxidation property of (0.5TiO2-Pt)/CeO2.

Comment 4: Section 2.2, the discussions about performance and should be revised, and combined with previous characterization result.

Response: Thanks for reviewer’s question. We have tested the reusability of (0.5TiO2-Pt)/CeO2 and concluded that the catalytic performance of (0.5TiO2-Pt)/CeO2 keeps stable after three repeated tests (Figure S5), implying good reusability of (0.5TiO2-Pt)/CeO2. The catalytic activity of (0.5TiO2-Pt)/CeO2 also surpasses the state of art TiO2/CeO2-based catalysts reported in literature, as shown in Table S1, indicating its superior catalytic property.

Comment 5: Fig. 8b, was that a DFT result? Did any characterization and result can support this figure?

Response: Thanks for reviewer’s question. Fig. 8 (Fig. 9 after revision) is not a DFT result, it is a possible mechanism for reaction of CO on (TiO2-Pt)/CeO2 catalyst. The reaction mechanism is concluded based on experiment and published literatures. Hopefully you will be satisfied with our explanation.

We tried our best to improve the manuscript and made some changes in the manuscript. These changes will not influence the content and frame work of the manuscript. And here we did not list the changes but marked in blue in the revised. We appreciate Editors/Reviewers’ warm work earnestly, and hope that the correction will meet with approval. Once again, thank you very much for your comments and suggestions.  Best wishes,

 Sincerely yours!

Reviewer 3 Report

Comments and Suggestions for Authors

Molecules

Manuscript ID:      molecules-2198344

Title:                            Enhanced catalytic performance of CeO2 supported TiO2-Pt nanorod in CO oxidation

In this manuscript, nanorod structured (TiO2-Pt)/CeO2 catalysts with addition of 0.3 at% Pt and different atomic ratio of Ti were prepared through a combined dealloying and calcination method. The catalysts were characterized by XRD, XPS, SEM, TEM and STEM in order to investigate the phase composition, surface morphology and structure of synthesized samples. I think that is an interesting experimental study, however, some issues should be improved. I recommend a publication of this work to Molecules journal after the authors consider the following major revisions.

Comment #1

The authors need to improve on their bibliography. I think that they should add some references in order to enrich the introduction section. In particular, the authors should improve the section of CeO2 in terms of the addition of other promoters into the structure, such as La3+, Sm3+ and or basic oxides in order to improve the physicochemical characteristics (basic sites, oxygen vacancies, oxygen mobility).

1.    Y. Zhang, J. Xu, X. Xu, R. Xi, Y. Liu, X. Fang, X. Wang Tailoring La2Ce2O7 catalysts for low temperature oxidative coupling of methane by optimizing the preparation methods. Catal. Today, 355 (2020), pp. 518-528,

2.    G.I. Siakavelas, N.D. Charisiou, A. AlKhoori, S. AlKhoori, V. Sebastian, S.J. Hinder, M.A. Baker, I.V. Yentekakis, K. Polychronopoulou, M.A. Goula, Highly selective and stable Ni/La-M (M=Sm, Pr, and Mg)-CeO2 catalysts for CO2 methanation. Journal of CO2 Utilization 51 (2021) 101618.

Comment #2

The authors need to be clearer on the motivation behind their work. Which is the innovation and what are the new aspects being introduced on this research topic? Please improve the introduction section

Comment #3

Which characterization technique the authors used for the measurement of Ni and promoters’ content?

Comment #4

The authors should correlate their catalytic performance results with the already published studies of different researchers to show the priority of their research study.

Comment #5

It possible for the authors to calculate kinetic data such as TOF or activation energy values?

Author Response

Dear Reviewer:

Thank you for your comments concerning our manuscript entitled “Enhanced catalytic performance of CeO2 supported TiO2-Pt nanorod in CO oxidation” (ID: molecules-2198344). Those comments are valuable and very helpful for revising and improving the manuscript, as well as the important guiding significance to our researches. According to the reviewers’ detailed suggestions, we have made a careful revision on the original manuscript. The revised are marked in yellow in the manuscript. The main corrections and the responds to the reviewers’ comments are as follows.

Response to reviewers’ comments:

Referee #4

Comment 1: The authors need to improve on their bibliography. I think that they should add some references in order to enrich the introduction section. In particular, the authors should improve the section of CeO2 in terms of the addition of other promoters into the structure, such as La3+, Sm3+ and or basic oxides in order to improve the physicochemical characteristics (basic sites, oxygen vacancies, oxygen mobility).

  1. Y. Zhang, J. Xu, X. Xu, R. Xi, Y. Liu, X. Fang, X. Wang Tailoring La2Ce2O7 catalysts for low temperature oxidative coupling of methane by optimizing the preparation methods. Catal. Today, 355 (2020), pp. 518-528,
  2. G.I. Siakavelas, N.D. Charisiou, A. AlKhoori, S. AlKhoori, V. Sebastian, S.J. Hinder, M.A. Baker, I.V. Yentekakis, K. Polychronopoulou, M.A. Goula, Highly selective and stable Ni/La-M (M=Sm, Pr, and Mg)-CeO2 catalysts for CO2 methanation. Journal of CO2 Utilization 51 (2021) 101618.

Response: Thanks for reviewer’s kind suggestion very much. We have revised introduction part carefully. We mentioned the challenges of current catalytic materials and pointed the highlights or strong points of our current works. We have also referred the sate of art literatures in the manuscript, see ref. 16, 17.

Comment 2: The authors need to be clearer on the motivation behind their work. Which is the innovation and what are the new aspects being introduced on this research topic? Please improve the introduction section

Response: Thanks for reviewer’s question. For one thing, Both theoretical and experimental studies have demonstrated that combing transition metal-oxides or rare   with noble metals is an effective method to reduce cost while maintaining catalytic property stable, which has been widely used for fuel cell and energy conversion/storage equipment. For another, the conventional fabrication methods always require relative high cost, complicated or time-consuming preparation process, which limit their large-scale application. Herein, by rational adjusting the proportion of TiO2 in the system, the obtained (0.5TiO2-Pt)/CeO2 displays unique nanorod structure and large pore volume, which contributes to exceptional catalytic activity. We think this work provides a new idea for preparation of high catalytic performance transition metal/CeO2-based catalysts for large-scale production.

Comment 3: Which characterization technique the authors used for the measurement of Ni and promoters’ content.

Response: Thanks for reviewer’s question. The content of Al, Ce, Pt, Ti in (0.5TiO2-Pd)/CeO2 catalyst obtained from Al91.2Ce8Pt0.3Ti0.5 melt-spun ribbon is 3.81 at%, 90.14 at%, 1.66 at%, 4.4 at%, respectively, as shown from the EDS spectrum in Fig.S1.

Comment 4: The authors should correlate their catalytic performance results with the already published studies of different researchers to show the priority of their research study

Response: Thanks for reviewer’s question. The catalytic activity of (0.5TiO2-Pt)/CeO2 also surpasses the state of art TiO2/CeO2-based catalysts reported in literature, as shown in Table S1, indicating its superior catalytic property.

Comment 5: It possible for the authors to calculate kinetic data such as TOF or activation energy values

Response: Thanks for reviewer’s question. We have added the characterizations about the catalytic activity of (0.5TiO2-Pt)/CeO2 under different space velocities as well as the catalytic performance under varied oxygen concentrations, see Fig. 8. The catalytic performance of (0.5TiO2-Pt)/CeO2 as a function of flow rate at 70 °C is detected, with corresponding shown in Fig. 8a. As the total gas flow rate increases from 40 to 120 ml min-1, the CO conversion decreases from 97% to 58%. It can be also clearly detected that the reaction rate is positive related to flow rate. Fig. 8b further explores the influence of O2 concentration in feed gas on catalytic performance of (0.5TiO2-Pt)/CeO2. The test temperature is kept at 90 °C with flow rate100 mL min-1. The CO conversion rate can reach 99% as 10% O2 is initially infused into the system thanks to the sufficient O2 environment; CO conversion rate is reduced first and then kept stable at 10% when O2 supply is suddenly decreased to zero, which may be ascribed to the existence of surface lattice oxygen that can migrate to active sites and combine with adsorbed CO to form oxygen vacancies. However, CO conversion rate increases in poor oxygen condition (0.3-5% O2) and then recovers to initial 99% value and keeps unchanged when O2 is resupplied into feed gas, implying superior catalytic CO oxidation property of (0.5TiO2-Pt)/CeO2.

We tried our best to improve the manuscript and made some changes in the manuscript. These changes will not influence the content and frame work of the manuscript. And here we did not list the changes but marked in blue in the revised. We appreciate Editors/Reviewers’ warm work earnestly, and hope that the correction will meet with approval. Once again, thank you very much for your comments and suggestions.  Best wishes,

 Sincerely yours!

Round 2

Reviewer 1 Report

Comments and Suggestions for Authors

Dear Editor
The paper was revised according to the reviewer’ comments.
In its current state it is ready for publication in your journal.
Best regards

Reviewer 2 Report

Comments and Suggestions for Authors

accept

Reviewer 3 Report

Comments and Suggestions for Authors

-